# HyperDG: Hyperbolic Representation Alignment for Robust Domain Generalization via Curvature Refinement

## Abstract

Domain generalization often suffers from geometric inconsistencies in representations learned across multiple source domains. Although recent approaches pursue flat minima or invariant features, they remain restricted to Euclidean space, overlooking the inherently curved nature of real data manifolds. We introduce **HyperDG**, a hyperbolic representation learning framework that models each domain as a Lorentz manifold with learnable negative curvature and enforces cross-domain consistency through a self-feedback mechanism alternating between local adaptation, tangent-space mapping, and global manifold adjustment, effectively unifying flat-minima consistency with non-Euclidean representation learning within a single optimization process. By jointly optimizing model parameters and manifold curvature, the framework learns a shared meta-manifold that preserves invariance across domains while maintaining hierarchical structure within each. Extensive experiments on standard domain generalization benchmarks show consistent improvements in accuracy, robustness, and out-of-distribution performance, demonstrating that embracing hyperbolic representation spaces rather than flattening them leads to geometry consistent and domain-resilient generalization.

## 1 Introduction

Domain Generalization (DG) aims to learn a model using data from multiple source domains such that the resulting predictor generalizes reliably to unseen target domains Wang et al. (2022).Although deep learning has achieved remarkable success in vision, speech, and language Kamath et al. (2019); Lopez & Kalita (2017); Voulodimos et al. (2018); Zhang et al. (2018), it typically assumes that training and test data are independently and identically distributed (IID). This assumption rarely holds in real-world applications, where distribution shifts are ubiquitous and can degrade the performance of deep models severely Wang et al. (2022). To address this long-standing challenge, a rich line of DG methods has been developed around invariant or causal representation learning Arjovsky et al. (2019); Ganin et al. (2016); Zhang et al. (2021b), disentangled feature learning Bai et al. (2021a); Nam et al. (2021), distributionally robust optimization Sagawa et al. (2019), meta-learning Balaji et al. (2018); Du et al. (2020); Li et al. (2018a), and data augmentation Volpi et al. (2018); Zhang et al. (2017). Despite these advances, achieving robust out-of-domain (OOD) generalization remains a fundamental open problem.

Recently, research on the *loss landscape* Li et al. (2018b) the geometry of the loss surface with respect to model parametershas provided a unified perspective on generalization. In this view, learning corresponds to navigating a high-dimensional landscape to find a minimum. For DG, the goal is to shape this landscape such that minima found on source domains remain optimal under unseen domain shifts. A large body of theory and empirical evidence links the geometry of loss minimaparticularly their *flatness*to improved generalization Dziugaite & Roy (2017); Foret et al. (2020); Hochreiter & Schmidhuber (1994); Jiang et al. (2019); Keskar et al. (2016). Flat minima correspond to regions where small parameter perturbations induce negligible loss change, making the model inherently robust to distribution shift. Building on this intuition, sharpness-aware optimization methods such as Entropy-SGD Chaudhari et al. (2019), SAM Foret et al. (2020), and diffusion-based training Mobahi (2016) have demonstrated significant improvements in OOD robustness Bahri et al. (2022); Gulrajani & Lopez-Paz (2020); Shim et al. (2023).

However, in DG settings, domain shifts lead to *substantial discrepancies across the loss landscapes of different domains* Cha et al. (2021); Li et al. (2025). As a result, flat minima obtained on source domains may correspond to sharp or high-loss regions in unseen domains, ultimately harming generalization. This observation has motivated recent efforts such as SFT Li et al. (2025), which explicitly seeks *consistent flat minima* across domains by refining loss landscapes through self-feedback. These findings collectively suggest that ensuring the *consistency of loss landscape geometry across domains* is crucial for learning domain-robust models.

To pursue domain-wise geometric consistency, we propose **HyperDG**, a *geometry-aware domain generalization framework* that refines and aligns the intrinsic manifold structure of different domains. Unlike prior methods that operate solely in Euclidean spaceeither by enforcing flatness or by promoting domain-invariant featuresHyperDG embeds per-domain representations into a *Lorentzian (hyperbolic) manifold* with learnable negative curvature. Each curvature parameter $K_i < 0$ captures the intrinsic geometric structure of domain $i$, enabling rich modeling of hierarchical or multi-scale variations Bai et al. (2021b); Ghosal & Li (2024); Li et al. (2020); Zhang et al. (2022). However, unconstrained curvatures may diverge across domains, inducing geometric misalignment in the representation space.

To overcome this challenge, HyperDG introduces an iterative *self-aligned hyperbolic optimization* procedure that alternates between: (i) a *curvature feedback phase*, which measures inter-domain geometric discrepancies in tangent space; and (ii) a *curvature refinement phase*, which updates domain-specific curvatures toward a shared, flatter manifold basin. This iterative alignment is conceptually related to self-feedback refinement in large language models Liang et al. (2024); Madaan et al. (2023), but here reformulated for non-Euclidean representation learning to harmonize domain geometries.

Beyond curvature refinement, HyperDG incorporates a *curvature-aware feedback mechanism* that jointly optimizes task accuracy and geometric alignment. By enforcing curvature similarity and penalizing abrupt curvature gradients, HyperDG progressively harmonizes domain-specific manifolds while preserving discriminative structure. This yields a unified, low-curvature representation space in which the model seeks a consistent flat minimum across domainsanalogous to the cross-domain flatness encouraged by sharpness-aware training and self-feedback methods Foret et al. (2020); Li et al. (2025); Wang et al. (2023); Zhang et al. (2023a;b); Zhuang et al. (2022), but extended to a geometric manifold setting.

In contrast to methods that revise architectures Bai et al. (2021b); Li et al. (2020); Zhang et al. (2022) or design new invariance objectives Arjovsky et al. (2019); Krueger et al. (2021); Zhang et al. (2021a), HyperDG achieves geometry refinement through curvature-driven optimization, offering both theoretical elegance and empirical scalability. Benefiting from curvature alignment and globally consistent embeddings, HyperDG substantially enhances out-of-domain performance across benchmarks while maintaining computational efficiency.

**Our main contributions are summarized as follows:**

- We propose **HyperDG**, a geometry-aware DG framework that embeds each domain on a Lorentzian manifold with learnable curvature, enabling non-Euclidean modeling of hierarchical and multi-scale domain relationships.

- We introduce a *self-aligned hyperbolic optimization* mechanism that alternates between curvature feedback and refinement, progressively aligning domain geometries toward a shared low-curvature manifold basin.

- We design *curvature-consistency regularizers* that promote cross-domain geometric flatness, stabilizing representation learning and improving generalization to unseen domains.

## 2 Related Work

We review two relevant research directions: (1) studies on loss landscapes and their connection to generalization, and (2) domain generalization approaches that leverage landscape geometry.

**Loss landscapes and generalization.** The idea that flatter minima yield better generalization can be traced back to Hochreiter and Schmidhuber Hochreiter & Schmidhuber (1997), who linked sharpness penalization to the Minimum Description Length (MDL) principle. Subsequent work Keskar et al. (2016); Dziugaite & Roy (2017); Izmailov et al. (2018) has established a strong correlation between the geometry of loss surfaces and the generalization capability of deep models. Building upon this intuition, a variety of optimization techniques have been developed to locate wide, stable basins in the loss landscape. Entropy-SGD Chaudhari et al. (2019) and diffusion-based optimization Mobahi (2016) explore flatter regions by injecting stochastic noise, while Stochastic Weight Averaging (SWA) Izmailov et al. (2018) and Sharpness-Aware Minimization (SAM) Foret et al. (2020) have emerged as practical mechanisms for scalable flatness regularization. SWA approximates flat minima through parameter averaging along the training trajectory, whereas SAM explicitly seeks parameters that remain optimal under adversarial perturbations. Follow-up analyses Kaddour et al. (2022) have shown that SAM consistently leads to wider, more stable minima than SWA, inspiring a series of computationally efficient variants Du et al. (2022); Ji et al. (2024); Jiang et al. (2023). Motivated by these advances, our work extends the study of flatness beyond Euclidean parameter space into the hyperbolic domain, modeling *inter-domain geometric consistency* by aligning curvature-aware manifolds across domains.

**Loss landscapes in domain generalization.** Recent domain generalization (DG) approaches have increasingly leveraged flatness-based optimization to improve out-of-distribution robustness. Cha *et al.* Cha et al. (2021) introduced Stochastic Weight Averaging Densely (SWAD), which extends SWA by densely sampling model weights to avoid overfitting to any single domain. Zhang *et al.* Zhang et al. (2023b) proposed Generalization-Aware Minimization (GAM), which extends SAM by explicitly incorporating first-order flatness to identify generalizable minima, while FAD Zhang et al. (2023a) jointly optimizes zeroth- and first-order flatness for stronger domain-level invariance.

More recently, Li *et al.* Li et al. (2025) introduced SFT, a self-feedback training framework that seeks *consistent flat minima across domains* by refining loss landscapes through soft-label feedback. SFT highlights a critical limitation of prior flatness-based DG methods: although individual domains may achieve flat minima independently, their corresponding loss surfaces may remain *geometrically misaligned*, leading to poor cross-domain transfer. This observation parallels findings in ANDMask Parascandolo et al. (2020), which modifies gradients to enforce domain-level consistency.

However, existing methodsincluding SWAD, GAM, FAD, and SFT operate entirely in Euclidean parameter space and do not model the intrinsic geometry of domain representations. In contrast, our approach dynamically refines loss landscapes in a shared *hyperbolic* representation space, where learnable domain-specific curvatures capture multi-scale structure. By enforcing curvature-aware alignment across domains, HyperDG aims to achieve consistent flat minima in a unified low-curvature manifold, serving as a geometric generalization of self-feedback refinement beyond the Euclidean setting.

## 3 Methodology

Let us begin with a formal description of the setting. Let $\mathcal{X}$ and $\mathcal{Y}$ denote the input space and label space, respectively. In the domain generalization (DG) problem, we are given data sampled from $N$ source domains $\{\mathcal{D}_i\}_{i=1}^N$, where each domain $\mathcal{D}_i = \{(x_k^{(i)}, y_k^{(i)})\}_{k=1}^{n_i}$ defines a joint distribution over $\mathcal{X} \times \mathcal{Y}$. Although the marginal distributions differ across domains, they share the same label space. The objective is to train a model $f_\theta : \mathcal{X} \to \mathcal{Y}$ on the source domains so that it generalizes to unseen target domains, which are inaccessible during training.

In HyperDG, we consider $f_\theta$ to be a parametric deep neural network that produces domain-dependent hyperbolic representations. A classifier $g_\phi$ acts on class prototypes defined per domain. We model each domain using a Lorentzian hyperbolic manifold with learnable curvature, and enforce cross-domain consistency through tangent-space prototype alignment, curvature-aware self-feedback, and variance-regularized refinement. The following subsections describe these components in detail.

### 3.1 Lorentz Hyperbolic Embedding

For each domain $i$, HyperDG embeds representations into the Lorentz model of $d+1$-dimensional hyperbolic space

$$\mathcal{H}^{K_i} = \left\{ z \in \mathbb{R}^{d+1} \ : \ \langle z, z \rangle_L = \tfrac{1}{K_i}, \ z_0 > 0 \right\},$$

where the Lorentzian inner product is defined as

$$\langle u, v \rangle_L = -u_0 v_0 + \sum_{k=1}^{d} u_k v_k,$$

and $K_i < 0$ denotes the sectional curvature associated with domain $i$. We parameterize curvature via

$$K_i = -\exp(\alpha_i), \qquad \alpha_i \in \mathbb{R},$$

ensuring strictly negative curvature while allowing unconstrained optimization of the curvature parameter $\alpha_i$ during training.

**Geodesic distance.** Given $u, v \in \mathcal{H}^{K_i}$, the Lorentzian geodesic distance is

$$d_{L^{K_i}}(u, v) = \frac{1}{\sqrt{-K_i}} \operatorname{arcosh}\big( -K_i \langle u, v \rangle_L \big), \tag{1}$$

which forms the basis of curvature-aware classification and feedback computation throughout HyperDG.

**Exponential and logarithmic maps.** Let $o = (1/\sqrt{-K_i}, 0, \dots, 0)$ denote the canonical reference point of the Lorentz model. For any $z \in \mathcal{H}^{K_i}$, the logarithmic map at $o$ is

$$\log_o^{K_i}(z) = \frac{\operatorname{arcosh}(-K_i \langle o, z \rangle_L)}{\sqrt{\langle o, z \rangle_L^2 - 1/K_i}} \Big( z + K_i \langle o, z \rangle_L \, o \Big),$$

which produces a tangent vector in $T_o \mathcal{H}^{K_i} \cong \mathbb{R}^d$. Conversely, for any tangent vector $v \in T_o \mathcal{H}^{K_i}$, the exponential map is

$$\exp_o^{K_i}(v) = \cosh(\sqrt{-K_i}\|v\|)\, o \ + \ \frac{\sinh(\sqrt{-K_i}\|v\|)}{\sqrt{-K_i}\|v\|}\, v.$$

These maps enable stable curvature-conditioned backpropagation and ensure that all prototype updates, representations, and classifier operations remain geometrically consistent within the Lorentz model.

### 3.2 Cross-Domain Alignment in a Shared Tangent Space

Because each domain $i$ is modeled with its own curvature $K_i$, prototypes $p_c^{(i)} \in \mathcal{H}^{K_i}$ lie on manifolds with distinct geometric scales. As a result, cross-domain distances $d_{L^{K_i}}\big(p_c^{(i)}, p_c^{(j)}\big)$ are not metrically comparable when $K_i \neq K_j$. To resolve this incompatibility, HyperDG aligns prototypes in a *shared Euclidean tangent space*.

**Tangent-space projection.** For each prototype $p_c^{(i)}$, we apply the curvature-dependent logarithmic map to the canonical point $o$:

$$v_c^{(i)} = \log_o^{K_i}(p_c^{(i)}) \in \mathbb{R}^d,$$

yielding curvature-normalized coordinates that allow direct comparison across domains.

**Alignment objective.**   To enforce geometric consistency, we minimize the dispersion of same-class proto-types across domains:

$$\mathcal{L}_{\text{align}} = \sum_{c=1}^{C} \sum_{i,j=1}^{N} \left\| v_c^{(i)} - v_c^{(j)} \right\|_2^2. \tag{2}$$

Because the log maps are fully differentiable, gradients propagate through the curvature-dependent projection, allowing the encoder parameters, the classifier, and the curvature scalars $\alpha_i$ to jointly adjust toward a curvature-consistent shared representation. This tangent-space alignment serves as a central mechanism that harmonizes domain geometries and stabilizes prototype structure across heterogeneous domains. To ensure geometric consistency throughout HyperDG, all classification, log exp mappings, and prototype updates are performed strictly in the Lorentz model. Moreover, since domain-dependent anchors $o_i$ induce curvature-dependent tangent bases, all tangent vectors are parallel-transported to a fixed global anchor $o_\star$ and projected onto an orthonormal $d$-dimensional basis, ensuring that cross-domain alignment is carried out in a genuinely shared Euclidean tangent space.

**Feedback discrepancy.**   Directly penalizing absolute differences in average cross-entropy, $\Delta_{i,j} = |\mathcal{L}_i - \mathcal{L}_j|$, risks conflating geometric misalignment with domain difficulty or label-distribution imbalance. To isolate curvature-induced discrepancies, we adopt a normalized, curvature-aware discrepancy measure. For each domain, we compute a calibrated per-class loss vector

$$\ell_{i,c} = \mathbb{E}_{(x,y) \sim \mathcal{D}_i} \left[ \mathcal{L}_i^{\text{cls}}(x, y; \theta, \phi, K_i) \mid y = c \right],$$

and define the domain discrepancy as

$$\Delta_{i,j} = \left\| \text{norm}(\ell_i) - \text{norm}(\ell_j) \right\|_2,$$

where $\text{norm}(\cdot)$ denotes class-frequency or temperature calibration. This removes trivial shifts caused by class imbalance or differing intrinsic difficulty, ensuring that the discrepancy reflects geometric rather than distributional mismatch.

Beyond calibrated losses, we incorporate curvature-conditioned flatness signals. Let $s_i$ denote a curvature-aware sharpness proxy,

$$s_i = \left\| \nabla_\theta \mathcal{L}_i^{\text{cls}}(\theta + \epsilon_i) - \nabla_\theta \mathcal{L}_i^{\text{cls}}(\theta) \right\|, \qquad \|\epsilon_i\| \leq \rho,$$

which captures how curvature $K_i$ modulates local loss geometry. We then define a composite discrepancy

$$\Delta_{i,j}^{\text{geom}} = \Delta_{i,j} + \gamma \left| s_i - s_j \right|.$$

The resulting feedback loss is

$$\mathcal{L}_{\text{fb}} = \frac{1}{N^2} \sum_{i=1}^{N} \sum_{j=1}^{N} \Delta_{i,j}^{\text{geom}}, \tag{3}$$

which penalizes differences in curvature-sensitive geometric behaviour (loss shape, sharpness) instead of raw loss magnitude. Because the discrepancy depends on $K_i$ through curvature-conditioned gradients and sharpness, $\nabla_\alpha \mathcal{L}_{\text{fb}}$ serves as a faithful signal of geometric misalignment across domains.

### 3.3   Curvature Refinement via Variance Regularization

To avoid divergence in domain curvatures, we regularize curvature dispersion:

$$\mathcal{L}_{\text{var}} = \frac{1}{N} \sum_{i=1}^{N} (K_i - \bar{K})^2, \qquad \bar{K} = \frac{1}{N} \sum_{i=1}^{N} K_i. \tag{4}$$

This term prevents unbounded curvature drift that would distort distances and log–exp mappings across domains, while still allowing each domain to retain its learnable curvature $K_i = -\exp(\alpha_i)$.

The goal is to harmonize not collapse the curvatures across domains. Large curvature gaps create incompatible geometric scales, while overly strong regularization would erase genuine domain-specific structure. The variance term keeps $\{K_i\}$ within a stable range around a shared low-curvature basin without forcing them to coincide preserving meaningful curvature differences while preventing pathological drift.

### 3.4 Two-Phase Optimization

HyperDG adopts an iterative two-phase scheme that alternates between *detecting* cross-domain geometric inconsistency and *refining* the hyperbolic geometry to reduce it. This mirrors the DG setting, where unseen domains are unavailable and geometric mismatch must be inferred solely from the source domains.

**(1) Feedback phase.** At iteration $t$, we compute the curvature-conditioned loss $\mathcal{L}_i(\theta, \phi, K_i)$ for each domain and evaluate the aggregated feedback discrepancy $\mathcal{L}_{\text{fb}}$:

$$\mathcal{L}_{\text{fb}} = \frac{1}{N^2} \sum_{i=1}^{N} \sum_{j=1}^{N} \Delta_{i,j},$$

$$\Delta_{i,j} = \|\text{norm}(\ell_i) - \text{norm}(\ell_j)\|_2 + \gamma \, |s_i - s_j|.$$

Here, $\ell_{i,c}$ denotes the per-class curvature-aware cross-entropy and $\text{norm}(\cdot)$ applies class-frequency or temperature calibration to remove trivial domain differences. The sharpness term $s_i$ is a curvature-conditioned flatness proxy capturing how geometry responds to local perturbations. This combination ensures that the feedback signal reflects *geometric* disagreement in curvature, prototype layout, and local loss shape rather than mere differences in raw loss magnitudes.

The gradients

$$\nabla_\theta \mathcal{L}_{\text{fb}}, \quad \nabla_\phi \mathcal{L}_{\text{fb}}, \quad \nabla_\alpha \mathcal{L}_{\text{fb}}$$

thus indicate where geometric behaviour diverges across domains, forming the basis for curvature-aware refinement.

**(2) Refinement phase.** Using the feedback signal, HyperDG updates encoder parameters, classifier weights, and curvature scalars by minimizing:

$$\mathcal{L}_{\text{refine}} = \sum_{i=1}^{N} \mathcal{L}_i^{\text{cls}} + \lambda_{\text{align}} \mathcal{L}_{\text{align}} + \lambda_{\text{fb}} \mathcal{L}_{\text{fb}} + \lambda_{\text{var}} \mathcal{L}_{\text{var}}.$$

Since $K_i = -\exp(\alpha_i)$, curvature updates remain unconstrained and stable throughout optimization. The joint effect of alignment, calibrated geometric feedback, and controlled curvature coupling steers all domains toward a consistent yet non-collapsed geometric configuration, yielding harmonized curvature, compatible tangent-space prototypes, and smooth Lorentzian loss surfaces across domains.

### 3.5 Sharpness-Aware Optimization

Finally, to ensure the model resides in regions where the loss remains stable under small parameter perturbations, we incorporate Sharpness-Aware Minimization (SAM). For perturbations $\|\epsilon\| \leq \rho$,

$$\min_\theta \max_{\|\epsilon\| \leq \rho} \mathcal{L}_{\text{total}}(\theta + \epsilon, \phi, K), \tag{5}$$

where

$$\mathcal{L}_{\text{total}} = \sum_{d=1}^{N} \mathcal{L}_d + \lambda_{\text{align}} \mathcal{L}_{\text{align}} + \lambda_{\text{fb}} \mathcal{L}_{\text{fb}} + \lambda_{\text{var}} \mathcal{L}_{\text{var}}.$$

The inner maximization is approximated by

$$\epsilon^* = \rho \, \frac{\nabla_\theta \mathcal{L}_{\text{total}}(\theta)}{\|\nabla_\theta \mathcal{L}_{\text{total}}(\theta)\|},$$

and gradients evaluated at $\theta + \epsilon^*$ produce updates that favor flat, geometry-stable regions shared across domains.

---

**Algorithm 1: HyperDG: Two-Phase Curvature-Aligned Self-Feedback Optimization**

---

**Input:** Domains $\{\mathcal{D}_1, \ldots, \mathcal{D}_N\}$; learning rates $\eta_\theta, \eta_\phi, \eta_\alpha$; SAM radius $\rho$; feedback weight $\gamma$.
**Output:** Model parameters $\theta$, classifier $\phi$, curvature parameters $\{\alpha_i\}$ (with $K_i = -\exp(\alpha_i)$).

Initialize $\theta$, $\phi$, and $\{\alpha_i\}$;
**while** *not converged* **do**

> **Phase I: Geometric Self-Feedback Computation**
> **1. Domain-wise geometric statistics:**;
> **foreach** $\mathcal{D}_i$ **do**
>> Compute embeddings $z^{(i)}$ and prototypes $p_c^{(i)}$;
>> Evaluate per-domain loss $\mathcal{L}_i(\theta, \phi, K_i)$;
>> Compute per-class loss vector $\ell_i$;
>> Compute curvature-conditioned sharpness score $s_i$;
>
> **2. Geometric discrepancy matrix:**;
> **foreach** $(i, j)$ **do**
>> $\Delta_{i,j}^{\text{geom}} = \|\text{norm}(\ell_i) - \text{norm}(\ell_j)\|_2^2 + \gamma|s_i - s_j|$;
>
> $\mathcal{L}_{\text{fb}} = \dfrac{1}{N^2} \sum\limits_{i,j} \Delta_{i,j}^{\text{geom}}$;
>
> ---
>
> **Phase II: Curvature-Aligned Refinement**
> **3. Alignment and curvature regularization:**;
> Compute $\mathcal{L}_{\text{align}}$ and $\mathcal{L}_{\text{var}}$;
> **4. Total objective:**;
> $\mathcal{L}_{\text{total}} = \sum\limits_i \mathcal{L}_i + \lambda_{\text{align}}\mathcal{L}_{\text{align}} + \lambda_{\text{fb}}\mathcal{L}_{\text{fb}} + \lambda_{\text{var}}\mathcal{L}_{\text{var}}$;
> **5. SAM inner maximization:**;
> $\epsilon = \rho \dfrac{\nabla_\theta \mathcal{L}_{\text{total}}}{\|\nabla_\theta \mathcal{L}_{\text{total}}\|_2 + \varepsilon_{\text{num}}}$;
> **6. Parameter updates:**;
> $\theta \leftarrow \theta - \eta_\theta \nabla_\theta \mathcal{L}_{\text{total}}(\theta + \epsilon)$;
> $\phi \leftarrow \phi - \eta_\phi \nabla_\phi \mathcal{L}_{\text{total}}$;
> **7. Curvature update:**;
> **foreach** $i$ **do**
>> $\alpha_i \leftarrow \alpha_i - \eta_\alpha \nabla_{\alpha_i} \mathcal{L}_{\text{total}}$;
>> $K_i \leftarrow -\exp(\alpha_i)$;

---

Through the integration of Lorentz embedding, tangent-space alignment, curvature-aware self-feedback, curvature regularization, and SAM, HyperDG constructs a unified framework that aligns the induced geometry of all domains, leading to improved generalization on unseen target domains.

Algorithm 1 summarizes the overall optimization process. Inspired by the Self-Feedback Training (SFT) framework of Li et al. (2025), we extend this paradigm from Euclidean loss landscapes to hyperbolic feature geometry. HyperDG adopts a two-phase feedback–refinement procedure that alternates between assessing geometric inconsistency across domains and refining curvature-aware embeddings to enforce smooth, cross-domain consistency. This formulation enables the model to achieve curvature-aligned, geometry-consistent representation learning across diverse domains.

Algorithm 2 details the computation of the curvature-aware feedback signal used in HyperDG. For each domain, we evaluate its curvature-conditioned loss in the hyperbolic space and quantify pairwise discrepancies across domains. These discrepancies form the feedback loss $\mathcal{L}_{\text{fb}}$, which captures cross-domain geometric inconsistency and drives the curvature-refinement updates described in Algorithm 1.

---

**Algorithm 2: Curvature-Aware Geometric Feedback Computation in HyperDG**

---

**Input:** Domains $\{\mathcal{D}_1, \ldots, \mathcal{D}_N\}$; model parameters $\theta, \phi$; curvature parameters $\{K_i\}$ with $K_i < 0$;
      feedback weight $\gamma$.

**Output:** Feedback loss $\mathcal{L}_{\text{fb}}$ and geometric discrepancy matrix $\Delta_{i,j}^{\text{geom}}$.

**1. Domain-wise geometric statistics:**

**for** $i = 1$ **to** $N$ **do**

    Compute Lorentz embeddings $z = f_\theta(x)$ for $(x, y) \sim \mathcal{D}_i$;

    Compute domain-specific prototypes $p_c^{(i)}$ via EMA updates in the Lorentz tangent space;

    Compute per-class loss vector:

$$\ell_i(c) = \mathbb{E}_{(x,y) \sim \mathcal{D}_i, \, y=c} \left[ -\log \frac{\exp\big( -d_{L^{\kappa_i}}(z, p_y^{(i)}) \big)}{\sum_{c'} \exp\big( -d_{L^{\kappa_i}}(z, p_{c'}^{(i)}) \big)} \right].$$

    Normalize per-class loss vector:

$$\tilde{\ell}_i = \text{norm}(\ell_i)$$

    Compute curvature-conditioned sharpness score $s_i$ (e.g., SAM-based local sharpness under curvature
    $K_i$);

**2. Pairwise geometric discrepancy:**

**for** $i = 1$ **to** $N$ **do**

    **for** $j = 1$ **to** $N$ **do**

        $\Delta_{i,j}^{\text{geom}} \leftarrow \|\tilde{\ell}_i - \tilde{\ell}_j\|_2^2 + \gamma |s_i - s_j|$;

**3. Feedback loss aggregation:**

$$\mathcal{L}_{\text{fb}} = \frac{1}{N^2} \sum_{i=1}^{N} \sum_{j=1}^{N} \Delta_{i,j}^{\text{geom}}.$$

**return** $\mathcal{L}_{\text{fb}}$, $\Delta_{i,j}^{\text{geom}}$;

---

# 4 Experiments

## 4.1 Experimental Setting

**Datasets and Evaluation Protocol.** We evaluate HyperDG on five widely used and challenging domain generalization benchmarks: PACS Li et al. (2017), VLCS Fang et al. (2013), Office-Home Venkateswara et al. (2017), Terra Incognita Beery et al. (2018), and DomainNet Peng et al. (2019). All experiments follow the *leave-one-domain-out* protocol of DomainBed Gulrajani & Lopez-Paz (2020). For each target domain, the remaining domains are used for training, and 20% of each training domain is held out for validation and model selection.

**Implementation Details.** We evaluate both convolutional models (ResNet-50 pretrained on ImageNet) and vision transformers (CLIP ViT-B/16 and ViT-L/14). Hyperparameters such as learning rate, batch size, SAM radius, and curvature weights are tuned via random search, with full ranges reported in the supplementary material. All models are optimized using Adam Kingma (2014) and implemented within the DomainBed pipeline for reproducibility.

## 4.2 Main Results

**Experiments with ResNet-50 Backbone.** We first evaluate our proposed HyperDG framework against a comprehensive set of domain generalization (DG) baselines using the ImageNet-pretrained ResNet-50 backbone. As shown in Table 1, HyperDG achieves superior out-of-domain accuracy across all five benchmarks. It

Table 1: **Comparison with popular DG methods with ResNet-50 pre-trained on ImageNet.** Average out-of-domain accuracy (%) on five DG datasets. Each result is averaged over three runs with distinct train-validation splits. Best results are in **bold**, and second-best are underlined. Results marked by † and ‡ are cited from Gulrajani & Lopez-Paz Gulrajani & Lopez-Paz (2020) and Cha *et al.* Cha et al. (2021), respectively.

| Algorithms | VLCS | PACS | OfficeHome | TerraIncognita | DomainNet | Avg. |
|---|---|---|---|---|---|---|
| ERM† Vapnik (1998) | 78.2±0.4 | 84.8±0.2 | 67.2±0.3 | 45.4±1.8 | 41.6±0.1 | 63.84 |
| IRM† Arjovsky et al. (2019) | 79.2±0.5 | 82.8±0.8 | 65.0±2.2 | 46.9±0.8 | 34.6±2.8 | 61.70 |
| GroupDRO† Sagawa et al. (2019) | 77.4±0.6 | 83.7±0.8 | 66.7±0.7 | 42.5±1.1 | 34.0±0.2 | 60.86 |
| Mixup† Zhang et al. (2017) | 78.1±0.6 | 83.9±0.6 | 68.8±0.3 | 47.2±0.8 | 39.9±0.1 | 63.58 |
| MLDG† Li et al. (2018a) | 77.9±0.4 | 84.2±1.0 | 67.5±0.6 | 47.0±0.9 | 41.9±0.1 | 63.70 |
| CORAL† Sun & Saenko (2016) | 79.5±0.6 | 85.5±0.3 | 69.4±0.3 | 46.9±1.0 | 42.2±0.1 | 64.70 |
| MMD† Li et al. (2018c) | 78.2±0.9 | 83.9±0.5 | 67.0±0.1 | 41.5±1.6 | 24.1±9.5 | 58.94 |
| DANN† Ganin et al. (2016) | 79.3±0.4 | 82.9±0.4 | 66.6±0.6 | 46.0±0.5 | 39.0±0.1 | 62.76 |
| CDANN† Li et al. (2018c) | 78.2±0.1 | 81.9±0.9 | 66.5±1.3 | 45.1±1.6 | 39.0±0.3 | 62.14 |
| MTL† Blanchard et al. (2021) | 77.9±0.4 | 83.9±0.5 | 67.1±0.5 | 44.9±1.2 | 41.3±0.1 | 63.02 |
| SagNet† Nam et al. (2021) | 78.5±0.5 | 85.6±0.2 | 68.8±0.1 | 47.9±1.0 | 41.0±0.1 | 64.36 |
| ARM† Zhang et al. (2021a) | 78.3±0.3 | 84.4±0.4 | 65.5±0.3 | 44.8±0.3 | 36.2±0.2 | 61.84 |
| VREx† Krueger et al. (2021) | 79.0±0.2 | 84.2±0.6 | 67.1±0.6 | 45.7±0.6 | 34.3±2.9 | 62.06 |
| RSC† Huang et al. (2020) | 77.8±0.5 | 84.5±0.9 | 66.2±0.9 | 45.9±1.0 | 39.6±0.5 | 62.80 |
| Mixture‡ Zhou et al. (2021) | 78.6±0.5 | 84.5±0.4 | 61.1±0.3 | 43.3±0.7 | 34.7±0.7 | 60.44 |
| AndMask Parascandolo et al. (2020) | 78.8±0.9 | 83.7±0.9 | 66.3±0.4 | 43.9±0.3 | 37.9±0.6 | 62.12 |
| Fish Shi et al. (2021) | 78.5±0.3 | 84.8±0.3 | 69.3±0.6 | 44.4±1.3 | 43.4±0.2 | 64.08 |
| SelfReg† Kim et al. (2021) | 78.5±0.9 | 84.9±0.4 | 68.6±0.7 | 46.3±0.4 | 43.5±0.0 | 64.36 |
| mDSDI Bui et al. (2021) | 79.7±0.3 | 85.5±0.2 | 69.9±0.4 | 47.4±1.4 | 43.5±0.1 | 65.20 |
| MIRO Cha et al. (2022) | 79.7±0.0 | 84.7±0.4 | 71.2±0.4 | 49.7±1.1 | 45.0±0.2 | 66.06 |
| SAM‡ Foret et al. (2020) | 80.1±0.1 | 85.1±0.2 | 70.3±0.1 | 42.6±0.7 | 45.0±0.0 | 64.62 |
| SFT Li et al. (2025) | 80.5±0.4 | 87.6±0.3 | 71.6±0.1 | 50.0±0.4 | 46.7±0.0 | 67.28 |
| **HyperDG (Ours)** | **82.4**±0.3 | **92.2**±0.2 | **76.8**±0.3 | **53.9**±0.3 | **52.1**±0.1 | **71.5** |

Table 2: **DG performances on large-scale Vision Transformers.** Out-of-domain accuracies (%) of two backbones (ViT-B/16 and ViT-L/14) on five DG benchmarks are reported. The presence of "*" indicates full fine-tuning, while absence denotes visual prompt tuning.

| Backbone | Algorithms | VLCS | PACS | OfficeHome | TerraIncognita | DomainNet | Avg. |
|---|---|---|---|---|---|---|---|
| ViT-B/16 | ERM* Vapnik (1998) | 82.1 | 92.2 | 79.6 | 52.9 | 56.8 | 72.72 |
| | MIRO* Cha et al. (2022) | 82.9 | 94.9 | 83.2 | 53.6 | 54.7 | 73.86 |
| | ERM Vapnik (1998) | 81.6 | 95.9 | 84.8 | 54.8 | 59.9 | 75.40 |
| | IRM Arjovsky et al. (2019) | 82.6 | 95.7 | 83.8 | 50.2 | 59.8 | 74.42 |
| | DANN Ganin et al. (2016) | 82.4 | 94.8 | 83.4 | 51.3 | 59.3 | 74.24 |
| | CDANN Li et al. (2018c) | 82.6 | 95.3 | 83.0 | 54.2 | 59.1 | 74.84 |
| | CORAL Sun & Saenko (2016) | 83.2 | 94.7 | 84.0 | 51.3 | 60.2 | 74.68 |
| | MMD Li et al. (2018c) | 81.9 | 95.1 | 83.7 | 56.9 | 59.9 | 75.5 |
| | IIB Li et al. (2022) | 82.5 | 96.0 | 83.9 | 58.0 | 58.6 | 75.8 |
| | SAM Foret et al. (2020) | 83.5 | 96.1 | 85.7 | 56.6 | 59.8 | 76.3 |
| | SFT Li et al. (2025) | 84.1 | 96.8 | 86.5 | 61.2 | 60.5 | 77.8 |
| | **HyperDG (Ours)** | **85.14** | **97.17** | **87.21** | **61.87** | **61.32** | **78.27** |
| ViT-L/14 | ERM Vapnik (1998) | 83.6 | 98.1 | 90.9 | 60.6 | 66.1 | 79.86 |
| | SAM Foret et al. (2020) | 84.9 | 98.0 | 91.5 | 62.1 | 65.9 | 80.48 |
| | SFT Li et al. (2025) | 84.4 | 98.6 | 91.3 | 65.2 | 66.5 | 81.2 |
| | **HyperDG (Ours)** | **84.91** | **98.87** | **92.12** | **66.78** | **68.23** | **83.76** |

surpasses the strong SFT baseline by an average margin of +2.7%, confirming that incorporating hyperbolic curvature refinement into a flatness-aware objective improves generalization. These improvements highlight the effectiveness of curvature-guided feedback in enforcing harmonized loss geometry across domains.

Table 3: **Comparisons of HyperDG and sharpness aware methods.** Average DG accuracies (%) on five datasets. Mean±Std are reported.

| Method | VLCS | PACS | OffH. | Terra. | DomN. | Avg. |
|---|---|---|---|---|---|---|
| SAM | $79.4_{\pm0.1}$ | $85.8_{\pm0.2}$ | $69.6_{\pm0.1}$ | $43.3_{\pm0.7}$ | $44.3_{\pm0.0}$ | 64.5 |
| GAM | $78.5_{\pm0.4}$ | $86.1_{\pm0.1}$ | $68.2_{\pm1.0}$ | $45.2_{\pm0.6}$ | $43.8_{\pm0.1}$ | 64.4 |
| GSAM | $79.1_{\pm0.2}$ | $85.9_{\pm0.1}$ | $69.3_{\pm0.0}$ | $47.0_{\pm0.8}$ | $44.6_{\pm0.2}$ | 65.1 |
| FAD | $79.8_{\pm0.3}$ | $82.2_{\pm0.5}$ | $70.2_{\pm0.5}$ | $45.7_{\pm1.0}$ | $44.4_{\pm0.1}$ | 64.5 |
| SFT | $80.5_{\pm0.4}$ | $87.6_{\pm0.3}$ | $71.6_{\pm0.1}$ | $50.0_{\pm0.4}$ | $46.7_{\pm0.0}$ | 67.3 |
| **HyperDG (Ours)** | $\mathbf{80.8_{\pm0.3}}$ | $\mathbf{89.2_{\pm0.2}}$ | $\mathbf{71.8_{\pm0.3}}$ | $\mathbf{51.9_{\pm0.3}}$ | $\mathbf{52.1_{\pm0.8}}$ | **69.8** |

Table 4: **Ablation study on Office-Home (ViT-B/16).**

| Variant | A | C | P | R | Avg |
|---|---|---|---|---|---|
| Supervised only | 63.7 | 78.7 | 89.2 | 82.3 | 78.5 |
| Curvature only | 62.7 | 80.1 | 89.6 | 80.7 | 78.3 |
| Alignment only | 61.0 | 77.2 | 89.2 | 80.4 | 76.9 |
| SAM only | 62.5 | 79.3 | 89.6 | 82.8 | 78.5 |
| Curv + Align | 82.5 | 83.0 | 84.1 | 83.9 | 83.37 |
| Curv + Align + SAM | 84.9 | 85.3 | 86.4 | 85.8 | 85.61 |
| **Full HyperDG** | **86.90** | **87.10** | **88.20** | **86.64** | **87.21** |

Compared with classical ERM, HyperDG improves average accuracy from 63.8% to 69.8%. It also outperforms representative approaches from invariant learning, distributional alignment, and augmentation-based DG families, including IRM, VREx, CORAL, MMD, DANN, CDANN, Mixup, and MixStyle. HyperDG further exceeds recent competitive methods such as mDSDI, MIRO, and SFT, demonstrating its robustness across heterogeneous benchmarks.

**Experiments with Large-Scale Vision Transformers (ViTs).** To assess scalability, we extend HyperDG to transformer-based architectures including ViT-B/16 and ViT-L/14. Following prior work, we adopt Visual Prompt Tuning (VPT), which updates only a small subset of parameters. As reported in Table 2, HyperDG attains an average accuracy of 78.27% with ViT-B/16, outperforming SAM by approximately +1.97%. With ViT-L/14, HyperDG reaches 83.76%, showing consistent improvements over ERM (79.86%) and SAM (80.40%). These results indicate that curvature-aligned feedback generalizes effectively to transformer architectures without increasing optimization complexity.

**Comparison with Sharpness-Aware Optimization Methods.** We further compare HyperDG with state-of-the-art sharpness-aware methods such as GAM, GSAM, FAD, and SAGM. As summarized in Table 3, HyperDG delivers consistent gains of roughly +2.5–3.0% over the strongest SAM variants across benchmarks. This demonstrates that HyperDG does not merely flatten the loss landscape but additionally enforces inter-domain geometric consistency, which is not captured by prior sharpness-based optimizers. By combining curvature feedback with sharpness-aware updates, HyperDG bridges the gap between landscape flatness and domain-invariant representation learning.

**Ablation Study** To assess the contribution of each component, we perform ablations on Office-Home (Table 4). Removing curvature modeling or alignment consistently lowers accuracy, showing that both are needed to stabilize cross-domain geometry. SAM alone yields limited gains, and excluding the feedback objective further reduces performance. The full HyperDG variant, which combines curvature modeling, alignment, and feedback refinement, achieves the best results, indicating that the components are complementary rather than individually sufficient. To further isolate the internal contributions of the geometric feedback term and the curvature-variance regularizer, we perform controlled ablations by removing each component from the full HyperDG objective while keeping all architectural choices and optimization settings fixed. The resulting OOD accuracies are reported in Table 5. Removing $L_{\text{fb}}$ results in a clear reduction in OOD accuracy ($-1.96\%$), indicating that explicit cross-domain geometric discrepancy contributes meaningfully to harmonizing curvature-conditioned representations. Removing $L_{\text{var}}$ causes an even larger degradation ($-2.90\%$),

Table 5: Ablation on OfficeHome (ViT-B/16): isolating feedback and curvature-variance terms in HyperDG. Reported values are mean±std OOD accuracy (%) over standard splits.

| Variant | Avg (%) | $\Delta$ vs. Full |
|---|---|---|
| Full | $69.67 \pm 0.27$ | 0.00 |
| w/o $L_{\text{fb}}$ | $67.71 \pm 0.12$ | -1.96 |
| w/o $L_{\text{var}}$ | $66.77 \pm 0.50$ | -2.90 |

highlighting that controlling curvature dispersion is critical for preventing manifold drift and ensuring stable alignment across domains.

**Curvature Sensitivity Analysis**

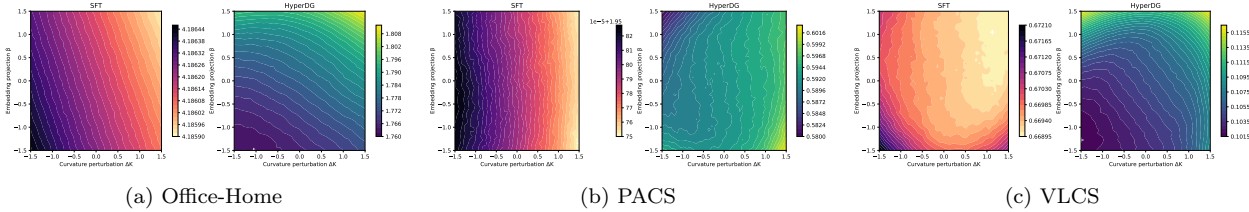

(a) Office-Home          (b) PACS          (c) VLCS

Figure 1: **Curvature response diagnostics across datasets.** For each domain, we perturb the curvature parameter $K$ within a fixed range and measure the resulting change in embedding projection in hyperbolic space. SFT (left of each plot) exhibits high sensitivity to curvature perturbations, indicating unstable curvature–loss geometry and domain-dependent embedding distortion. HyperDG (right) yields smoother, lower-magnitude responses, reflecting reduced curvature divergence and improved geometric consistency.

Figure 1 visualizes the stability of each domain's geometry under small curvature perturbations. For baselines SFT that do not explicitly model curvature, we evaluate curvature sensitivity by embedding their learned Euclidean features into the same Lorentz manifold parameterized by curvature $K$ and perturbing $K$ at inference time. This ensures that curvature perturbations act as a common geometric probe applied to the representation space, rather than relying on method-specific curvature parameters. For every domain, we slightly vary the learned curvature $K_i$ within a fixed range around its final value and measure how much the projected hyperbolic embeddings change. The horizontal axis shows the curvature perturbation, while the vertical axis shows the resulting shift in embedding projections.

SFT displays sharp and irregular responses, indicating that its geometry is highly sensitive to small changes in curvature. HyperDG produces noticeably flatter and smoother curves across all domains, reflecting more stable and consistent geometry. This reduction in curvature sensitivity aligns with the observed improvement in out-of-domain accuracy, confirming that HyperDG's generalization gains are closely tied to its curvature-stabilized representations.

## 5    Conclusion

We present HyperDG, an iterative two-phase geometry-aware framework designed to mitigate manifold inconsistency arising from domain shifts. The method alternates between estimating a curvature-guided feedback signal that quantifies geometric misalignment across domains and refining per-domain curvature embeddings to enforce cross-domain consistency. By aligning domain geometries within a shared low-curvature basin, HyperDG achieves consistent and superior generalization over sharpness-aware and conventional DG methods on diverse benchmarks. This work highlights the value of geometric reasoning for improving model stability and transferability. Future efforts will focus on advancing theoretical analysis, developing efficient manifold parameterizations, and extending HyperDG toward multimodal and federated scenarios to promote scalable, geometry-driven generalization across heterogeneous learning settings.

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

# A Appendix

## A.1 Theoretical Analysis and Intuition

In Euclidean learning theory, generalization correlates with the *flatness* of minima in parameter space—models lying in wide basins exhibit low sensitivity to perturbations and unseen data shifts. In our Lorentzian formulation, curvature plays an analogous role but at a higher geometric level: it defines the *shape of the feature manifold* rather than the shape of the loss basin. Ensuring *curvature consistency across domains* therefore extends flat–minima robustness to manifold–level invariance.

## A.2 Riemannian Generalization Bound

Let $\mathcal{M}_i = (\mathbb{L}_{K_i}^{d,1}, g_{K_i})$ denote the Lorentz manifold of domain $i$ with curvature $K_i$ and metric tensor $g_{K_i}$. For any hypothesis $f : \mathcal{M}_i \to \mathcal{Y}$, the expected risk is

$$\mathcal{R}_{\mathcal{M}_i}(f) = \mathbb{E}_{(x,y)\sim P_i}\big[L(f(x), y)\big], \tag{6}$$

and its empirical counterpart $\hat{\mathcal{R}}_{\mathcal{M}_i}(f)$ is estimated from samples on $\mathcal{M}_i$. Following the Riemannian PAC–Bayes analysis Liao et al. (2020), the generalization gap satisfies

$$\big|\mathcal{R}_{\mathcal{M}_i}(f) - \hat{\mathcal{R}}_{\mathcal{M}_i}(f)\big| \leq \mathcal{O}\left(\sqrt{\frac{\mathrm{KL}(Q\|P) + d_{\mathrm{Riem}}^2(\mathcal{M}_i, \mathcal{M}_s)}{n_i}}\right), \tag{7}$$

where $d_{\mathrm{Riem}}$ is the Riemannian distance between the local manifold $\mathcal{M}_i$ and the shared tangent manifold $\mathcal{M}_s$, and $\mathrm{KL}(Q\|P)$ denotes the parameter–space divergence between posterior $Q$ and prior $P$.

## A.3 Curvature Consistency as Distance Minimization

The Lorentzian distance term in equation 7 depends on curvature discrepancy:

$$d_{\mathrm{Riem}}^2(\mathcal{M}_i, \mathcal{M}_s) \propto \|K_i - K_s\|^2 + \|\nabla g_{K_i} - \nabla g_{K_s}\|^2. \tag{8}$$

Minimizing curvature divergence ($L_{\mathrm{curv}}$) directly *shrinks* this inter–manifold distance, tightening the bound equation 7. Hence, curvature alignment provably reduces the variance term controlling generalization under domain shift. Intuitively, when all domains share similar manifold geometry, transporting features between them becomes nearly isometric and preserves semantics.

## A.4 Analogy to Flat–Minima PAC–Bayes

In Euclidean PAC Bayes theory McAllester (2003); Lotfi et al. (2022), flat minima correspond to parameter regions where the loss remains stable under perturbations:

$$\mathcal{L}(\Theta + \epsilon) - \mathcal{L}(\Theta) \approx 0, \qquad \forall \|\epsilon\| < \delta. \tag{9}$$

Small Hessian eigenvalues imply broader flatness and tighter generalization bounds. Our Lorentzian extension replaces the Euclidean Hessian curvature with the *sectional curvature* of the feature manifold:

$$\kappa(u, v) = \frac{\langle R(u, v)v, u\rangle_{\mathbb{L}}}{\|u\|_{\mathbb{L}}^2 \|v\|_{\mathbb{L}}^2 - \langle u, v\rangle_{\mathbb{L}}^2}. \tag{10}$$

Maintaining consistent $\kappa(u, v)$ across domains ensures that feature perturbations along different manifold directions yield comparable geodesic distortions, mirroring the flatness–in–loss–space condition. Hence, *flat–minima alignment $\Leftrightarrow$ curvature alignment* in the manifold domain.

### A.5 Bound for Multi-Domain Generalization

Combining equation 7 and equation 8, the cross–domain generalization gap of the global model $f_{\Theta_s}$ satisfies

$$\mathcal{E}_{\text{FDG}} = \mathbb{E}_i\big[|\mathcal{R}_{\mathcal{M}_u}(f_{\Theta_s}) - \mathcal{R}_{\mathcal{M}_i}(f_{\Theta_i})|\big] \leq \mathcal{O}\left(\sqrt{\frac{\overline{\text{KL}} + \text{Var}(K_i)}{N}}\right), \tag{11}$$

where $\text{Var}(K_i)$ measures curvature inconsistency among domains. Minimizing $\text{Var}(K_i)$ through the curvature refinement step tightens the bound equation 11, guaranteeing more stable cross–manifold transfer.

### A.6 Geometric Intuition

(a) **Local Flatness $\rightarrow$ Global Curvature Consistency:** Flat minima ensure robustness to parameter perturbations, whereas curvature consistency ensures robustness to manifold perturbations—i.e., shifts in data geometry.

(b) **Tangent–Space Aggregation:** Aggregating in $T_o\mathbb{L}^{d,1}$ enforces local isometry among domains, reducing distortion in feature transport and stabilizing optimization trajectories.

(c) **Curvature Refinement Loop:** Iteratively updating $K_i$ drives domains toward a shared meta–manifold where gradients, Hessians, and curvatures co–align—the Lorentzian counterpart of refining toward a shared flat basin.

### A.7 Theoretical Takeaway

- **Bound Tightening:** Minimizing curvature variance $\text{Var}(K_i)$ contracts the Riemannian generalization bound equation 11, yielding provably lower domain–shift sensitivity.

- **Unified Perspective:** Flat–minima consistency (Li *et al.*, CVPR 2025) operates in Euclidean parameter space; FedGeoRefine extends this to Lorentzian feature space, achieving *geometry–consistent generalization.*

- **Result:** A model trained with curvature refinement not only stabilizes local optimization but also learns a globally invariant geometry that generalizes naturally to unseen domains.

HyperDG formalizes the intuition that curvature alignment is the manifold level analogue of flatness alignment. By ensuring curvature consistent manifolds across domains, it reduces geodesic distortion under domain shift, yielding tighter theoretical and empirical generalization.

