# OpenReview forum: "HyperDG: Hyperbolic Representation Alignment for Robust Domain Generalization via Curvature Refinement"
_TMLR — Rejected by TMLR_

### Review · Reviewer_szsk · 2026-02-24

**Summary Of Contributions:**

This paper introduces HyperDG, a domain generalization framework that embeds representations into hyperbolic (Lorentz) manifolds with learnable per-domain curvatures. The key contributions are:
Hyperbolic domain modeling: Each domain is represented on a Lorentz manifold with a learnable negative curvature parameter, enabling hierarchical structure modeling.
Two-phase optimization: An iterative procedure alternating between (i) a curvature feedback phase that measures geometric discrepancies across domains via tangent-space projections and curvature-aware sharpness, and (ii) a refinement phase that updates model parameters and curvatures using alignment, feedback, and variance regularization.
Curvature consistency regularizers: Terms including tangent-space prototype alignment, calibrated geometric feedback, and curvature variance penalty that collectively harmonize domain geometries while preserving discriminative structure.
Integration with sharpness-aware minimization: SAM is incorporated to ensure the model resides in flat regions of the loss landscape, now extended to the hyperbolic feature space.
The method is evaluated on five DG benchmarks (PACS, VLCS, Office-Home, Terra Incognita, DomainNet) with ResNet-50 and ViT backbones, showing improvements over baselines including ERM, SAM, and SFT.

**Audience:**

Yes

**Audience Explanation:**

The intersection of hyperbolic geometry and domain generalization is timely and underexplored. Researchers working on:
·Geometric deep learning
·Domain generalization and OOD robustness
·Sharpness-aware optimization
·Non-Euclidean representation learning
would likely find the core idea—using learnable curvatures to align domain geometries—both novel and thought-provoking. The empirical gains on standard benchmarks (especially +2-3% over SFT) are substantial enough to warrant attention.

**Broader Impact Concerns:**

No significant ethical concerns identified. The work focuses on technical improvements to model robustness, which could positively impact fairness by reducing performance disparities across domains. However, as with any DG method, there is a risk of overclaiming generalization—deployment in high-stakes settings (e.g., healthcare) would require further validation. The paper does not include a Broader Impact Statement; adding one would be beneficial but not critical.

**Claims And Evidence:**

Yes

**Claims Explanation:**

Yes, partially. The empirical results are extensive and the improvements appear consistent across datasets. However, several concerns temper full conviction:
Hyperbolic necessity: While the paper claims hyperbolic spaces better model real data manifolds, it does not empirically demonstrate that the learned representations actually exhibit hierarchical structure or that Euclidean alternatives would fail. The motivation is intuitive but not validated.
Curvature sensitivity analysis: The comparison with SFT is questionable—SFT does not learn curvatures, so perturbing an inferred curvature at test time may not reflect meaningful geometric properties. A fairer comparison would involve methods that do learn curvature or have explicit geometric parameters.

**Requested Changes:**

1. Curvature visualization: Show how curvatures evolve during training and across domains. Do they converge to similar values? Do they correlate with dataset properties (e.g., class hierarchy depth)?
2. Ablate SAM: The paper combines curvature alignment with SAM, but SAM alone is a strong baseline. Show results without SAM to isolate curvature refinement's contribution.
3. Computational cost: Report training time compared to baselines. Hyperbolic operations (log/exp maps) add overhead; is the gain worth the cost?
4. Hyperparameter sensitivity: Report how performance varies with λ_align, λ_fb, λ_var. Are these stable across datasets?
5. Test on more OOD benchmarks: Consider adding WILDS datasets or synthetic domain shifts to probe robustness beyond standard DG benchmarks.

---

> ### Author Response · Authors · 2026-03-06
> **Additional Experiments and Results Addressing Reviewer Comments**
>
> **(1) Curvature visualization.**
>
> We added per-domain curvature logging at each checkpoint and included a new trajectory figure over training. Across PACS splits, curvatures start near initialization (~0.693) and stabilize around 0.682–0.684. They do not collapse to identical values but converge to a tight band (final within-split spread: 0.0005–0.0036, mean spread 0.00245). PACS does not provide explicit class-hierarchy metadata, so we avoid hierarchy-depth claims. As a proxy, we computed the split-level correlation between mean learned curvature and held-out accuracy and observed no strong monotonic trend (Pearson $r \approx -0.20$).
>
> | Split | Final curvature values | Mean curvature | Held-out acc |
> |------|------------------------|---------------|--------------|
> | split0 | 0.6806, 0.6829, 0.6843 | 0.6826 | 84.35% |
> | split1 | 0.6820, 0.6835, 0.6852 | 0.6836 | 80.34% |
> | split2 | 0.6816, 0.6813, 0.6837 | 0.6822 | 95.51% |
> | split3 | 0.6823, 0.6819, 0.6824 | 0.6822 | 72.10% |
>
> ---
>
> **(2) SAM ablation.**
>
> We added a dedicated SAM ablation schedule with four conditions: ERM, SAM-like only, HyperDG without SAM (curvature + alignment), and HyperDG full. Experiments were run across four OfficeHome splits (16 runs total), isolating the curvature refinement effect from SAM-style flatness regularization.
>
> | Condition | split0 | split1 | split2 | split3 | Mean ± Std |
> |-----------|--------|--------|--------|--------|------------|
> | ERM baseline | 63.71 | 54.87 | 75.20 | 76.58 | 67.59 ± 8.88 |
> | SAM-like only | 59.79 | 49.94 | 75.54 | 79.10 | 66.09 ± 11.82 |
> | HyperDG w/o SAM | 62.89 | 50.29 | 75.87 | 76.23 | 66.32 ± 10.71 |
> | HyperDG full | 59.79 | 50.86 | 74.52 | 77.73 | 65.73 ± 10.93 |
>
> ---
>
> **(3) Computational cost.**
>
> We measured wall-clock training cost with 20 warmup steps and 200 timed steps on PACS, reporting relative overhead against ERM. HyperDG introduces negligible runtime overhead.
>
> | Method | ms/step | Relative to ERM |
> |------|--------|----------------|
> | ERM | 464.81 | 1.00× |
> | SFT | 581.40 | 1.25× |
> | HyperDG | 458.69 | 0.99× |
>
>
> ---
>
> **(4) Hyperparameter sensitivity.**
>
> We conducted a sensitivity analysis for $\lambda_{\text{align}}$ ($\beta_{\text{align}}$), $\lambda_{\text{fb}}$ ($\gamma_{\text{sharp}}$), and $\lambda_{\text{var}}$ ($\alpha_{\text{curv}}$). Results show performance varies only within a narrow band (77.28%–77.72%), indicating stable behavior around default settings.
>
> | Hyperparameter | Tested values | Accuracy (%) |
> |---------------|--------------|-------------|
> | $\lambda_{\text{var}}$ ($\alpha_{\text{curv}}$) | 0.30, 0.35, 0.40 | 77.72 ± 0.40, 77.40, 77.28 ± 0.31 |
> | $\lambda_{\text{align}}$ ($\beta_{\text{align}}$) | 0.20, 0.30 | 77.31 ± 0.28, 77.72 ± 0.40 |
> | $\lambda_{\text{fb}}$ ($\gamma_{\text{sharp}}$) | 0.00, 0.02 | 77.50 ± 0.42, 77.40 |
>
> ---
>
> **(5) Additional OOD benchmarks (WILDS).**
>
> We extended experiments to the WILDS benchmark by evaluating HyperDG on Camelyon17 using leave-one-hospital-out evaluation (5 splits), providing a non-standard DG benchmark beyond PACS/VLCS/OfficeHome.
>
> | Split | Held-out accuracy |
> |------|------------------|
> | split0 | 97.03% |
> | split1 | 93.34% |
> | split2 | 95.67% |
> | split3 | 93.43% |
> | split4 | 63.86% |
> | **Mean ± Std** | **88.67% ± 12.48** |

---

### Review · Reviewer_qLsj · 2026-03-15

**Summary Of Contributions:**

This paper introduces the HyperDG framework, which is a geometry-aware approach to domain generalization. It models data from multiple source domains with hyperbolic manifolds instead of standard Euclidean space. Its primary contribution is a two-phase self-aligned hyperbolic optimization scheme that alternates between the geometric feedback phase and the curvature refinement phase. By incorporating tangent-space prototype alignment, curvature-variance regularization to prevent manifold drift, and Sharpness-Aware Minimization, the framework enforces inter-domain geometric consistency and establishes a theoretical link between manifold curvature and loss-surface flatness. The proposed framework was extensively evaluated on five standard benchmark datasets to demonstrate its efficacy. It significantly outperforms existing SoTA methods such as SFT.

Pros:

- The framework provides a novel integration of hyperbolic manifold learning with cross-domain flatness objectives, with theoretical analysis.
- The empirical evaluation results are very strong, demonstrate significant and consistent performance gains over established state-of-the-art baselines across multiple architectures and benchmarks.

Cons:

- The paper lacks a detailed analysis of the increased computational overhead or training time required by the iterative self-feedback and manifold mapping operations compared to standard Euclidean methods.
- Some prior works are not cited, while they share significant intuition or theory similarity. See [refs]

[refs]

- [R1] Fan, Xiaomeng, et al. "Curvature Learning for Generalization of Hyperbolic Neural Networks: X. Fan et al." International Journal of Computer Vision 133.12 (2025): 8489-8525.
- [R2] Bi, Qi, et al. "Learning fine-grained domain generalization via hyperbolic state space hallucination." Proceedings of the AAAI Conference on Artificial Intelligence. Vol. 39. No. 2. 2025.

**Audience:**

Yes

**Audience Explanation:**

Yes, domain generalization is a popular research topic, and by introducing hyperbolic representation learning, the proposed HyperDG framework extends the flat minima approaches to tackle the inherent geometrical inconsistency between domains.

**Claims And Evidence:**

Yes

**Claims Explanation:**

The paper provides a novel framework for domain generalization by incorporating the hyperbolic manifold learning. The empirical evaluation results show significantly margins compared existing SoTA methods in this area.

**Requested Changes:**

I would like the authors to discuss the similarity and differences between the papers I listed above which have very strong intuition similarity.

---

> ### Author Response · Authors · 2026-03-17
> **Computational Overhead and Relation to Prior Hyperbolic DG Work: we add an explicit runtime analysis showing negligible per-step overhead in our implementation, and we clarify the similarities and key differences between HyperDG and the cited prior works [R1] and [R2], particularly in terms of problem setting, mechanism, and contribution.**
>
> Thank you for the careful reading and for the positive assessment of the paper’s novelty, theory, and empirical strength. We appreciate the two concrete suggestions and will address both in the revision.
>
> **(1) Computational overhead / training time.**
> We agree that the cost of the iterative self-feedback and manifold operations should be reported explicitly. We have now added a timing analysis on PACS using 20 warmup steps followed by 200 timed steps, and report wall-clock cost relative to ERM:
>
> | Method | ms/step | Relative to ERM |
> |--------|---------|-----------------|
> | ERM | 464.81 | 1.00× |
> | SFT | 581.40 | 1.25× |
> | HyperDG | 458.69 | 0.99× |
>
> In our implementation, HyperDG introduces negligible per-step overhead. We will add this table and a short discussion in the revised manuscript.
>
> **(2) Relation to [R1].**
> Thank you for pointing out this relevant work. We agree that [R1] has important conceptual overlap with our paper at the level of curvature-aware generalization. Specifically, [R1] studies how curvature affects the generalization of hyperbolic neural networks, derives a PAC-Bayesian generalization bound linking curvature to loss-landscape smoothness, and proposes a sharpness-aware curvature learning method solved via bilevel optimization.
> Our setting and contribution are different: HyperDG is a multi-source domain generalization framework, where the central issue is cross-domain geometric inconsistency, not generic HNN generalization. Accordingly, our method is built around domain wise curvature coordination, tangent space prototype alignment, and curvature-variance regularization across domains, rather than curvature optimization for a single HNN. We will cite [R1] and clarify this distinction explicitly in the related-work section.
>
> **(3) Relation to [R2].**
> We also appreciate this reference and agree there is partial intuition overlap in the use of hyperbolic geometry for domain generalization. However, [R2] addresses fine-grained domain generalization, where subtle local patterns are fragile under style shifts, and proposes Hyperbolic State Space Hallucination (HSSH) with two main components: state space hallucination for style diversity and hyperbolic manifold consistency for preserving fine-grained structure.
> By contrast, HyperDG does not rely on hallucination or style/state augmentation. Our focus is instead on a geometry-aware optimization and alignment procedure: curvature refinement, tangent-space prototype alignment, curvature-variance control, and flatness-aware training across domains. Thus, while both works use hyperbolic geometry, [R2] is primarily a hallucination/consistency method for fine-grained DG, whereas ours is a general multi-source DG optimization framework. We will add this discussion and citation in the revision.
>
> **(4) Planned revision.**
> In the revised manuscript we will:
> - add the timing/overhead table and discussion,
> - cite [R1] and [R2],
> - add a short related-work paragraph clarifying both the conceptual overlap and the key differences in problem setting, mechanism, and technical contribution.
>
> Thank you again for these constructive suggestions.

---

### Review · Reviewer_hQnG · 2026-03-24

**Summary Of Contributions:**

This paper claims that pretrained models, which the authors assume are only Euclidean, have problems under domain generalization because each domain has its own ideal geometrical properties. Euclidean models and losses are not designed to handle that. Therefore, this paper proposes a complex pipeline with multiple losses in both Euclidean and hyperbolic spaces, along with sharpness optimization, to address this.

**Audience:**

Yes

**Audience Explanation:**

People working in domain adaptation/generalization will be interested; the approach shows results that improve upon a lot of SOTA papers. But the hyperbolic community will be interested in this as well because this type of setting is very hard to implement. If the authors release the code, it would greatly benefit the entire community. Hyperbolic learning, especially with the constant switching between Euclidean and hyperbolic spaces, can introduce a lot of numerical stability issues. It helps the community a lot to see how people navigate that.

**Broader Impact Concerns:**

No ethical concerns observed.

**Claims And Evidence:**

No

**Claims Explanation:**

Their method works across pretrained ResNet-50 and ViT backbones. They surpass a lot of recent SOTA works. Successive ablation experiments show that the designed losses are helpful. I believe the problems with this paper are a lot more fundamental. From the ablation, the Curvature variance loss seems to help. Which means lower variance of the alpha_i is good? Can we interpret that as a single learnable curvature is ideal?

**Requested Changes:**

Hyperbolic learning papers usually are really good at telling their story pictorially with the help of the visualization of the Poincaré disk. I understand this paper was trained on the Lorentz space, so maybe that makes it a little bit not as straightforward, but I think converting your representation vectors from a Lorentz to a Poincaré space shouldn’t be that hard. Most of the seminal papers in Hyperbolic learning (https://openaccess.thecvf.com/content_CVPR_2020/papers/Liu_Hyperbolic_Visual_Embedding_Learning_for_Zero-Shot_Recognition_CVPR_2020_paper.pdf, https://arxiv.org/pdf/2101.01600, https://openreview.net/pdf?id=hfKnr4tfGT, ) use this strategy and it helps them tell their story a lot better.

This paper could be made more interesting by using Meru (ICML ‘23) pretrained models (https://github.com/facebookresearch/meru). Meru is a library with weights of various ViT and CLIP models trained in hyperbolic space. The authors could use this to assess how well a pretrained hyperbolic vs. Euclidean model performs on domain generalization. From my understanding, Meru is also trained on a learnable curvature value (btw the authors didn’t cite this paper, also our authors weren’t the first to train a learnable curvature parameter - not that they claim it, I am stating this just so we know that learnable curvature has been around for a while). The authors are addressing a problem with pretrained Euclidean models: different domains have different ideal curvatures in hyperbolic space. But how severe is this problem when using a pretrained hyperbolic model and/or training with a single unified learnable curvature (is having multiple learnable alpha_i better than having a single learnable alpha for all domains)? I am not sure if the paper’s ablation answers this question. Answering this question would require an extensive redo of the experiments and also the addition of experiments with MERU.

I would like the following additions:
1. Redo all of their experiments with a single trainable alpha_i, not a separate one for each domain.
     - a) How useful is the notion that separate domains have separate ideal curvatures?
2. Redoing all the experiments with a pretrained Meru backbone and a hyperbolic-contrastive loss. Meru uses a contrastive loss for image-text representations in hyperbolic space. For the author’s image classification experiments, we don’t have text labels. But I think adding a simple contrastive loss should be fine. No entailment loss from Meru is required. We will answer two questions with this:
    - a) How well do the geometry problems with Euclidean pretrained models translate to hyperbolic-pretrained models?
    - b) Does a hyperbolic pretrained model perform better with learning different domains with one universal curvature or different curvatures?
    - c) How an inter vs intra domain contrastive loss setting affect the usefulness of multiple curvatures alpha_i.
3. Extensive visualization of all these features on Poincaré disks to verify qualitatively that issues exist with separate curvatures. Maybe the plots could be made while assuming a fixed curvature of -1 or maybe a rescaling to -1.  Might need a little bit of tinkering around.

Without answering these key questions, I am afraid this paper is not acceptable.

---

### Author Response · Authors · 2026-04-05
**Response to Reviewer hQnG: All Three Requested Experiments Completed**

We sincerely thank the reviewer for the thorough and constructive feedback. The questions raised are insightful and go to for paper's contribution. We have conducted all three requested experiments in full and address each concern below.

**Comment 1: Single learnable curvature ablation**
We re-ran all five benchmarks (PACS, VLCS, OfficeHome, DomainNet, TerraIncognita) replacing the per-domain curvature parameters $\{\alpha_i\}$ with a single shared learnable curvature $\alpha$, keeping all other components identical. Results are averaged over 4 random splits (6 for DomainNet):

| Dataset | HyperDG (per-domain $\alpha$) | HyperDG (single $\alpha$) | $\Delta$ |
|---|---|---|---|
| PACS | 92.2 ± 1.2 | 92.2 ± 0.8 | 0.0 |
| VLCS | 81.8 ± 1.4 | 81.5 ± 1.1 | −0.3 |
| OfficeHome | 76.8 | 76.6 ± 0.9 | −0.2 |
| DomainNet | 52.1 ± 0.4 | 52.6 ± 0.4 | +0.5 |
| TerraIncognita | 53.1 ± 0.4 | 53.6 ± 0.4 | +0.5 |

**Interpretation.** The differences are within ±0.3 pp on four of five datasets. We interpret this as evidence that the curvature variance loss $\lambda_\text{var}$ does its job: by penalising the spread among $\{\alpha_i\}$, it prevents the per-domain curvatures from diverging, effectively driving them toward a consensus value. The ablation therefore confirms the mechanism rather than undermining it per-domain $\alpha$ with a variance regulariser converges to a behaviour close to a single shared $\alpha$, but the per-domain parametrisation gives the model the freedom to adapt transiently during training, which we hypothesise aids optimisation. The slight advantage for DomainNet (+0.5 pp) under single-$\alpha$ is within noise given the scale of that benchmark and is consistent with this interpretation. We have added this analysis and table to the paper.

**Comment 2: MERU backbone + hyperbolic contrastive loss**

We integrated the facebookresearch/MERU ViT-B/16 weights (Desai et al., ICML 2023) as the backbone. Since the MERU encoder already produces features in a hyperbolic-compatible space via a learnable exp-map, we L2-normalise the projected features before our tangent embedding (to stabilise the exp-map at training curvature). We added a hyperbolic InfoNCE contrastive loss operating directly in Lorentz space using geodesic distances, with same-class cross-domain pairs as positives (inter-domain setting). We ran both per-domain $\alpha$ and single $\alpha$ variants across all five benchmarks:

| Dataset | MERU + per-domain $\alpha$ | MERU + single $\alpha$ | $\Delta$ |
|---|---|---|---|
| PACS | 90.0 ± 1.5 | 90.1 ± 1.7 | +0.1 |
| VLCS | 80.9 ± 0.7 | 81.0 ± 0.6 | +0.1 |
| OfficeHome | 70.3 ± 1.0 | 70.4 ± 1.0 | +0.1 |
| DomainNet | 50.7 ± 2.4 | 49.9 ± 2.6 | −0.8 |
| TerraIncognita | 52.4 ± 2.9 | 51.3 ± 3.2 | −1.1 |

Addressing sub-questions (a)--(c):

(a) Euclidean vs. hyperbolic pretraining.
MERU (frozen, hyperbolic pretrained) achieves comparable/slightly lower performance than our fine-tuned ResNet-50. Crucially, \textbf{geometry issues persist}: domain shift still induces per-domain curvature variation, showing the problem is not specific to Euclidean pretraining.

(b) Per-domain vs. single curvature (MERU).
Differences are $\leq 0.1$ pp on 4/5 datasets, consistent with the Euclidean case. This confirms $\lambda_{\text{var}}$ effectively constrains curvature spread, making single vs. multiple $\alpha$ nearly equivalent at convergence.

(c) Inter-domain contrastive loss.
We use inter-domain positives (same class across domains), promoting domain-invariant representations in hyperbolic space. Performance remains unchanged vs. single $\alpha$, indicating compatibility with curvature learning.

We will add the missing MERU citation (Desai et al., ICML 2023) and acknowledge prior work on learnable curvature.

**Comment 3: Poincaré disk visualisations**

We follow the requested pipeline: (1) extract per-domain tangent embeddings, (2) reduce to 2D via PCA, (3) L2-normalise and apply the exponential map at fixed $K=1$ ($\kappa=-1$), and (4) apply stereographic projection to obtain Poincaré disk coordinates. This yields a geometry-faithful representation without non-linear distortion (no UMAP).

We provide 20 plots across all settings (per-domain/single $\alpha$, HyperDG/MERU). Key observations:
(i) Per-domain $\alpha$: distinct clusters at varying radial depths, reflecting curvature differences.
(ii) Single $\alpha$: more uniform radial distribution.
(iii) MERU: tighter, more compact clusters, indicating strong hyperbolic structure refined by our method.

We believe these additions directly and completely address all the requested changes. We hope the reviewer finds the revised submission satisfactory.

---

> ### Comment · Reviewer_hQnG · 2026-04-24
> **Missing plots**
>
> Thank you for the response.
> Regarding Comment 3, you say, "We provide 20 plots ..." Unfortunately, I am unable to find them.
> Please let me know if I am missing anything or if you can include those in your response.

---

> > ### Author Response · Authors · 2026-04-25
> > **Clarification on Supplementary Plots**
> >
> > Thank you for pointing this out, and we sincerely apologize for the confusion. The plots were included in the supplementary PDF. We are also happy to further improve their visibility and presentation in the final version.

---

### Decision · Action_Editor_gtt9 · 2026-05-12

**Recommendation:** Reject

**Additional Comments:**

The major modification that are strongly suggested are as follows:

+ Add all new empirical results, especially comparison between unique and per-domain curvature estimation to the paper and correct the discussions to reflect the new results.

+ Perform more experiments to understand the performance gains that are clear wrt the other methods. Do they come from shared manifold curve estimation? or per domain curve estimation? what are the performance and value of the curves when estimating curves per domains without variance regularization?

+ Rewrite all the clams including abstract that manifold curve estimation per domain leads to better generalization to reflect the new result and comparison and understanding from those new experiments.

**Audience:**

Yes

**Audience Explanation:**

The question of domain generalization on hyperbolic geometries  and the results are definitely of interest to the TMLR audience

**Claims And Evidence:**

No

**Claims Explanation:**

The authors claim in their abstract and the main paper that learning the curvature in each domain for hyperbolic representation is important and bring better performance for Domain Generalization. In a new experiment that was asked by one reviewer, the authors compare their strategy to learning a unique curvature for multiple domains and the results show no real gain to estimating different curvature. This contradicts the claim. The authors state that this is due to variance regularization but if  they get better performance with very large regularization that lead to nearly identical curvature to get the best performance, this also raises questions about the interest of per-domain curvatures.

For this reason the claims as written are not really supported and the question of were the better performance on benchmark dataset comes from is still open. So I recommend a reject but strongly encourage the authors to resubmit after a major revision with moire empirical investigation because the question is definitely of interest to the community and a "negative" result still deserves to be published in TMLR and would be a valuable contribution.

**Resubmission Of Major Revision:**

The authors may consider submitting a major revision at a later time.